# The Sixth Mass Extinction and Amphibian Species Sustainability Through Reproduction and Advanced Biotechnologies, Biobanking of Germplasm and Somatic Cells, and Conservation Breeding Programs (RBCs)

**DOI:** 10.3390/ani14233395

**Published:** 2024-11-25

**Authors:** Robert K. Browne, Qinghua Luo, Pei Wang, Nabil Mansour, Svetlana A. Kaurova, Edith N. Gakhova, Natalia V. Shishova, Victor K. Uteshev, Ludmila I. Kramarova, Govindappa Venu, Mikhail F. Bagaturov, Somaye Vaissi, Pouria Heshmatzad, Peter Janzen, Aleona Swegen, Julie Strand, Dale McGinnity

**Affiliations:** 1Sustainability America, Sarteneja, Corozal District, Belize 91011, Belize; 2Hunan Engineering Technology Research Center for Amphibian and Reptile Resource Protection and Product Processing, College of Biological and Chemical Engineering, Changsha University, Changsha 410022, China; dani2017@126.com; 3Hunan Engineering Laboratory for Chinese Giant Salamander’s Resource Protection and Comprehensive Utilization, School of Biological Resources and Environmental Sciences, Jishou University, Jishou 416000, China; wangpei0229@126.com; 4Fujairah Research Centre, University of Science and Technology of Fujairah, Fujairah P.O. Box 2202, United Arab Emirates; nabil.mansour@frc.ae; 5Institute of Cell Biophysics, Russian Academy of Sciences, PSCBR RAS, Pushchino 142290, Moscow Region, Russia; sakaurova@mail.ru (S.A.K.); gakhova@gmail.com (E.N.G.); cryopreservation@list.ru (N.V.S.);; 6Institute of Theoretical and Experimental Biophysics, Russian Academy of Sciences, Pushchino 142290, Moscow Region, Russia; luda_kramarova@rambler.ru; 7Centre for Applied Genetics, Department of Zoology, Jnana Bharathi Campus, Bangalore University, Bengaluru 560056, Karnataka, India; venugcaecilian@gmail.com; 8IUCN/SSC/Athens Institute for Education and Research/Zoological Institute RAS, St. Petersburg 199034, Northern Region, Russia; bbigmojo@mail.ru; 9Leningrad Zoo, St. Petersburg 197198, Northern Region, Russia; 10Department of Biology, Faculty of Science, Razi University, Baghabrisham, Kermanshah 57146, Iran; s.vaissi@razi.ac.ir (S.V.); pheshmatzad@gmail.com (P.H.); 11Department of Fisheries, Faculty of Fisheries and Environmental Sciences, Gorgan University of Agricultural Sciences and Natural Resources, Gorgan 49138, Iran; 12Justus-von-Liebig-Schule, 47166 Duisburg, Germany; pjanzen@gmx.de; 13School of Environmental and Life Sciences, College of Engineering, Science and Environment, University of Newcastle, Callaghan, NSW 2308, Australia; aleona.swegen@newcastle.edu.au; 14Department of Chemistry and Bioscience, Aalborg University, Fredrik Bajers Vej 7K, 9220 Aalborg Ost, Denmark and Randers Regnskov, Torvebryggen 11, 8900 Randers C, Denmark; js@biosfaeren.dk; 15Ectotherm Department, Nashville Zoo at Grassmere, Nashville, TN 37211, USA; dmcginnity@nashvillezoo.org

**Keywords:** mass extinction, COP 16, COP 28, reproduction technologies, intergenerational justice, de-extinction, climate catastrophe, assisted evolution, terraforming, space colonization

## Abstract

Primary themes in intergenerational justice are a healthy environment, the perpetuation of Earth’s biodiversity, and the sustainable management of the biosphere. These goals demand transformative changes to biodiversity management, especially when considering the predicted sixth mass extinction. Reproduction and advanced biotechnologies, biobanks of germplasm and somatic cells, and conservation breeding programs (RBCs) provide a transformative change to perpetuate biodiversity irrespective of environmental targets, ecosystem collapses, and other sixth mass extinction drivers. Future potentials for RBCs include assisted evolution, species restoration, and the extension of the biosphere through interplanetary and interstellar colonization. We address these themes with amphibian models to introduce the MDPI Special Issue, The Sixth Mass Extinction and Species Sustainability through Reproduction and Advanced Biotechnologies, Biobanking, and Conservation Breeding Programs.

## 1. Introduction

The Earth’s biosphere is under threat from landscape modification, which, along with catastrophic global heating and other threats, is driving profound changes in ecosystems globally and accelerating the sixth mass extinction [1]. COP 15, 2022, ”Ecological Civilisation: Building a Shared Future for All Life on Earth”, and COP 16, 2024, “Sixteenth meeting of the Conference of the Parties to the Convention on Biological Diversity”, focus on direct habitat protection [2,3]. However, the COP 15 targets of protecting 30% of the world’s land and water and restoring 30% of degraded ecosystems by 2030 are unlikely to be met, with only 12 countries contributing only USD 413 million toward the 200 billion per year needed. Even if COP 15 targets were met, habitat modification, even in protected habitats, driven by catastrophic global heating, exotic predators and competitors, emergent pathogens, and major losses in prey and other taxa [3,4], will inevitably result in the mass extinction of many amphibian species in the wild [5,6,7,8]. Amphibians are the most threatened vertebrate group, where, as of the present date, the IUCN accesses 8011 species, with 36.4% (2912) having some degree of threat of extinction, with 16.8% (1263) species being Endangered and 10.0% (799) species Critically Endangered, and with these most threatened categories comprising ~23.5% of the 8772 described species on AmphibiaWeb. Only 52.3% of IUCN species are Near Threatened or of Least Concern, with 11.3% (908) being Data Deficient. Of further concern is that ~65% of populations are decreasing, with only ~40% being stable [9,10,11]. Similarly, rapid declines are also occurring across all terrestrial vertebrates [12].

COP 28, 2023 [13], emphasized the urgency of addressing catastrophic global heating as a major factor in forcing the sixth mass extinction. Catastrophic global temperature increases from pre-industrial levels, based on Intergovernmental Panel on Climate Change (IPCC) estimates, are conservative. For 2023, the IPPC predicted a 1.10 °C increase [1], whereas the actual increase was 1.46 °C [14], and that for 2040 is 1.40 °C [1], with alternative models predicting at least 1.85 °C [14,15]. IPPC underestimates could result from projection models, including a reduction of CO_2_ emissions and increasing CO_2_ capture, and disregarding increased heating through crossing global climate tipping points [14,15]. Meeting CO_2_ emission-reduction targets has failed for most countries, with some countries increasing emissions [16], and prohibitively costly amelioration projections depending on speculative technologies for CO_2_ removal are unlikely to be achieved within realistic timeframes [16,17,18].

Even more alarming are increasing emissions of CO_2_ from disastrous forest fires, rapidly increasing their role as a carbon source rather than a sink [19,20,21], increasing seawater temperatures and acidity, reduced oceanic CO_2_ absorption [22,23], and predictions of the potential collapse by the mid to late 21st century of the Atlantic Meridional Overturning Circulation (AMOC) oceanic current, which affects weather patterns globally, including possible season reversals in highly amphibian-biodiverse major bioregions [24].

Plausibly, more than the current 2062 Endangered or Critically Endangered amphibian species will eventually become extinct in the wild because of catastrophic global heating synergized with other threats [25,26]. Global heating will particularly affect amphibians subject to altitudinal habitat constraints [26,27,28], dependent on permanent stream flows or wetlands [29], or sensitive to forest-habitat destruction through fires [19,20,21,30,31,32] or elevated temperatures beneath tropical forest canopies [33], with predictions that by 2080–2100 up to 35% of anuran habitats will become arid or exposed to worsening drought [34]. Furthermore, global heating will synergize with other environmental stressors, including endocrine-disrupting chemicals [35], micro/nano plastic pollution [36], ice and permafrost melt [37], and the release of toxic chemicals such as endocrine disruptors and heavy metals including mercury and iron [36,38], that will likely be occurring at a much faster rate than can be offset by the slow processes of artificially augmented natural reproduction, resulting in even greater impacts on amphibians and other biodiversity and ecosystems [39]. For example, bird populations have halved in large and otherwise considered pristine Ecuadorian rainforest reserves [40], and insects and other invertebrates have experienced major population declines globally [41]. Because biodiversity is irreplaceable, and its loss so functionally and ethically catastrophic, the precautionary principle directs that we should take the direst global heating predictions as our baseline for an immediate and emphatic response, including the general adoption of reproduction and advanced biotechnologies, biobanking of germplasm and somatic cells, and conservation breeding programs (RBCs) for species perpetuation [42].

However, governments have not taken seriously the needs of intergenerational justice that entitle future generations to a healthy environment [43], sustainable biospheric management that includes the perpetuation of Earth’s biodiversity [3,44,45,46,47], and especially addressing catastrophic global heating [48,49]. Therefore, transformative change and supportive biotechnical, political, and cultural initiatives [50,51] are needed to reduce or prevent biodiversity loss and ameliorate the sixth mass extinction [49,52,53,54].

These transformative approaches must embrace biotechnological advances rather than relying on optimistic stopgap, adversarial, and traditionalist approaches [3,8,50,51,55,56]. Transformative approaches are slowly gaining traction through the International Geosphere–Biosphere Programme [57], the United Nations Convention on Biodiversity [58], the International Union for the Conservation of Nature (IUCN) One Plan Approach to Conservation [59,60], the 2024 Amphibian Conservation Action Plan [61], Amphibian Ark [62,63], and private caregiver conservation breeding programs (CBPs) [3,64,65]. Furthermore, in 2021 the One Conservation concept for species conservation emphatically promoted “reproductive biotechniques” for species conservation [66]. These transformative approaches to varying extents included interventional RBC strategies [3] and the promotion of positive societal changes [67]. Futuristic approaches include species perpetuation through long-term cell and tissue storage in biobanks that, for security, could eventually be secured extra-terrestrially [68]. However, the 2024 Amphibian Conservation Action Plan (ACAP) was not inclusive and disregarded both the enormous potential of private caregiver CBPs [65] and the perpetuation of species through germplasm and somatic-cell biobanking [3].

The application of RBCs offers a transformative change that, irrespective of environmental targets, can: (1) reliably and economically maintain genetic diversity within CBPs (Section 2. [10]) and produce genetically adaptable individuals for repopulation, augmentation, or supplementation programs [3,10]; (2) integrate ex situ and in situ conservation within a broader cultural context, bringing all layers of society as agents of conservation [66,69,70,71,72,73,74,75,76]; and (3) provide the potential for species perpetuation solely in biobanks [3,68,77,78,79,80,81,82,83]. These interventional activities can also benefit humanity through cultural and social inclusion, economic and technological advancement, and educational opportunities [3,67,84]. Overall, RBCs offer a significant advancement in the management of biospheric sustainability and the avoidance of mass extinctions by providing a safety net for the perpetuation of Endangered and Critically Endangered species [3,66,69,70,71,72,73,74,75,76,82,85,86,87]. Sustainability interventions, including amphibian RBCs, have mainly focused on anurans (frogs and toads) and salamanders. Caecilians (Gymnophiona) are also an amphibian order of special concern. Besides a general disregard for caecilians in the public and scientific arena [88], other threats include many data-deficient species of unknown conservation status, a lack of autecological knowledge because of their generally subterranean habitats [89], and little research supporting the development of caecilian RBCs [66,71,90,91,92].

Two decades have passed since the potentials of amphibian reproduction and advanced biotechnologies were first published in Australia in 1999 [70], fertilization with cryopreserved amphibian sperm was developed in Russia [93,94], its use for fertilization in Australia [95,96], and heterocytoplasmic cloning in Japan in 1963 [97,98]. Despite these pioneering achievements, amphibian RBCs have only been implemented for a few species, primarily in wealthy western polities [99,100] and through their satellite conservation programs in the global south [101].

This geopolitical limitation creates a critical gap in amphibian conservation, as very-high-biodiversity countries in the global south are not provided with the necessary independent resources to develop the full potential of RBCs. The development of biobanks representative of all Critically Endangered and Endangered species globally is also particularly lacking [3]. Nevertheless, preventing the immediate extinction of some amphibian species through RBCs has already been achieved. Between 2007 and 2017, there was an increase in amphibian CBPs of ~60%, covering 2.9% of all amphibian species. Half of these CBPs involved repopulation or augmentation, and 70% of these had some success [10,61,102,103,104]. RBCs using sperm biotechnologies have also benefitted the linking of in situ and ex situ conservation programs [99,100]

Cultural factors can play a major role in the widespread adoption of RBCs. Scientific publications and public media should utilize an international, clear, and concise nomenclature system that empowers public engagement and optimizes search-engine visibility [3]. At an international level, this approach, along with oversight beyond the scientific review process, avoids regionalized jargon, misnomers, and euphemisms that hinder the powerful and accurate scientifically based dissemination of information [3,105,106,107]. These initiatives are now even extending to addressing historical injustices in species nomenclature [108]. Pioneering achievements should be properly attributed to promoting global cooperation in a multipolar world (Section 3.7 and Section 4.1).

Philosophical and ethical considerations concerning amphibian animal welfare in RBCs have been particularly prominent with respect to the collection of cells or tissues (Section 5), the release of individuals into potentially hostile ecosystems for repopulation or assisted gene flow (Section 2.2, [103,104]), and advanced reproduction biotechnologies, including assisted evolution or species restoration (Section 4). Despite these ethical considerations (Section 5), intergenerational justice and looming sixth mass extinctions [25,26] demand the prioritization of biobanking of germplasm and somatic cells to enable the opportunist restoration of species through cloning or other advanced techniques (Section 4). Furthermore, to ensure the perpetuation of amphibian species through RBCs, a concerted effort toward international democratic engagement within the emerging multipolar world is essential. This collaboration should broadly include local communities, environmental organizations, and private caregivers’ amphibian collections and their organization’s caregivers [3,64,66,109,110,111,112,113].

In summary, this review explores the potential of RBCs by exemplifying amphibians through (Section 2) management models for genetic diversity; (Section 3) an overview of RBC protocols including their utility, potential, limitations, and research; (Section 4) advanced reproduction biotechnologies, including cloning and assisted evolution; and (Section 5) the ethical and philosophy aspects of RBCs to foster public trust and support. Finally, in Section 6, we summarize current and future applications of amphibian RBCs, particularly the fostering of community engagement and international collaboration to secure a future rich in amphibian biodiversity for the foreseeable future [3]. Overall, this review presents a hopeful outlook for the perpetuation of amphibian biodiversity through embracing current and novel biotechnologies, along with fostering international cooperation through exemplary management and social and cultural sensitivities.

## 2. Genetic Diversity and Species Management

Reduced genetic diversity can lead to genotypes with reduced potential for environmental adaptation, poor health, and lowered fecundity [114,115,116,117,118,119,120,121,122,123,124,125]. A goal of the IUCN’s One Plan Approach to Conservation is the maintenance of sufficient genetic diversity to provide fitness and evolutionary adaptability. It includes all agencies operating in concert to perpetuate species as both wild and captive populations or biobanked genetic resources of individuals [59,60] in a single genetic management unit called a metapopulation [126]. Therefore, a structured approach using genetic databases of both living amphibians and biobanked material combined with genetic modeling is essential for a scientifically credible RBC program [81,82,99,117,119,120,121,122,127].

Capturing the maximum genetic diversity for ex situ genetic management [81,99,114,115,116,117,128,129] includes the number and proportions of living male, female [120], and biobanked founders [73,76,129]. The IUCN Amphibian Population Management Guidelines recommend 25 founder pairs to represent 99.5% of the target population’s genetic diversity (Figure 1, left panel, [119,120]). An equal number of founder males and females is preferable; however, a bias toward live females would maximize progeny production for release [130].

The number of founders and initial allelic frequency can be used to calculate the likelihood that an allele will persist in future generations (Figure 1, right panel, [92]). However, a greater challenge is the quantification of an allele’s potential for future environmental adaptation. For example, a rare allele might be crucial for developing resistance to a new disease [131] or providing camouflage in a habitat with dramatic seasonality in the chromaticism of vegetation structure [132]. For example, the green and golden bell frog, *Litoria aurea*, a sun-basking species, inhabits wetlands subject to seasonal changes in vegetation color from green to brown, with *L. aurea* varying genotypically and consistently chromatically throughout an individual’s lifespan from pure gold to green, with most individuals being a mixture (Figure 2).

Besides the IUCN Population Management Guidelines [119], alternative approaches have recommended different numbers of biobanked founder individuals and their sex ratios (Table 1, ibid).

Genetic management of mating in CBPs is crucial for preventing inbreeding between closely related individuals [136]. Depending on the species-generation time and lifespan, with an initial population of 25 live males and females, preserving 90% of genetic diversity in a CBP for 25 years requires the maintenance of a minimum number of 100 individuals with strict studbook pairing and 1600 individuals with group management and random mating (Table 2, [120]). Unfortunately, despite careful studbook pairing to track the breeding history of individuals, unexpected deaths, domestic adaptations, and epigenetic effects, can result in the loss of genetic diversity and allelic variation (Figure 1) and progeny poorly adapted to survival in the wild [116,123,124,137].

A key role of the biobanking of germplasm, somatic cells, or tissues is to increase the effective population size beyond the limited number of breeding individuals typically available in CBPs. Sperm storage and artificial fertilization, particularly utilizing banked cryopreserved sperm, are the only current biotechnologies contributing to the prevention of the gradual loss of genetic diversity in amphibian CBPs [71,73,75,76,138]. Biobanked sperm pauses the genetic aging process within a CBP and provides a reservoir of genetic variation, including rare alleles that might be crucial for future adaptation to environmental changes such as disease resistance [131], modified ecosystems [4,6], or predators [132].

The benefits of biobanking sperm are exemplified by real-world examples. Case studies involving three frog species demonstrated how using biobanked sperm for breeding (back-crossing) reduces the required size of the CBP’s population, minimizes inbreeding, and lowers costs when compared to maintaining large numbers of individuals [139,140]. Biobanked sperm can restore genetic diversity to highly inbred color morphs, irrespective of their genetic diversity, in private caregiver collections, thus avoiding costly intuitional CBPs altogether and encouraging private caregivers’ engagement in species perpetuation. For instance, the popular, attractive, and highly endangered *Atelopus* sp. (Figure 3) could potentially be perpetuated through private caregiver collections along with genetic diversity provided through cryopreserved sperm [64,141]).

Some publications have recommended the biobanking of germplasm and, to a lesser extent somatic cells, of all threatened amphibian species, Vulnerable, Endangered, and Critically Endangered [128,133]. However, because of limited resources, biobanking should primarily focus on the immediate needs of the current ~1630 Endangered and particularly Critically Endangered species [9,10,11]. Vulnerable species are not in immediate danger of extinction [9,11,142], and their management should emphasize autecological IUCN recommendations for fieldwork, research and monitoring, and particularly habitat protection, including encouraging community engagement [3,9,11,142]. Nevertheless, if resources are available, opportunistic biobanking of some Vulnerable species close to endangerment could be undertaken before the possible loss of genetic diversity with major population declines occurs.

### 2.1. Genetic Bias in Founder Sperm Collection

Minimizing anthropocentric, terminal-investment effect and founder selection bias, ensures the optimal representation of the target population’s genetic diversity [75,76]. Anthropogenic bias occurs through collectors favoring traits such as vocalization [143], antagonism and larger size [144,145], and chromaticism [146]. However, these traits may not represent a species’ adaptive genetic diversity [147,148,149], as they are weakly associated with mating systems, habitat types, and life history [145]. Although age is associated with size and senescence theory predicts loss of sperm quality, and therefore yield, no effect was found by Watt et al. [150].

Evolutionary trait constraints are possible through increased antagonistic inter-male competitive traits [144,145] corresponding to decreased female mate choice, along with ‘sneaker’ males and delayed partial-clutch fertilization influencing the success of fertilization [151]. Further research is needed to fully reveal the relationships between sperm genetic diversity [152,153], female sperm choice in fertilization [154], and the fitness of resultant progeny [155].

Biases could also be due to genetically predisposed characteristics toward pathogens, including susceptibility to more overt parasite-mediated behavior [156], greater size corresponding to the intensity of infection [157], or through terminal investment effects where sick or potentially dying individuals invest larger resources than usual to increase reproductive output [158]. Terminal investment effect was evident through increased spermatogenesis of males from two anuran families during hormonal stimulation when infected by the major-threat fungal pathogen of amphibians [158]. Furthermore, infections with parasites [157] from taxonomic groups with Endangered species [159,160,161] also increased spermatogenesis in one species [158,160]. However, in another species, male advertisement and mating outcomes were lower in fungal-pathogen-affected males [162]. Besides potential bias toward parasites, sperm collection through hormonal stimulation includes the possibility of lower sperm yield from less-mature or more stress-prone males (See Section 4.2. [163]).

### 2.2. Assisted Gene Flow

Assisted gene flow (AGF), where desirable genotypes are transferred between CBPs, biobanks, and wild populations to reduce inbreeding depression, is an emerging RBC that could potentially increase population fitness but also entails considerable risks [10,75,99,102,164,165,166,167]. Reduced inbreeding depression should increase reproductive capacity, evolutionary adaptability, and consequently, species survivability [114,115,116,117,118,119,120,121,122,123,124,128,142,168,169,170]. However, these advantages are dependent on the target population’s size, its unique genetic diversity, and whether it is a metapopulation, including its fragmented sub-populations, or a genetically unique population isolated over geological periods [117,171,172]. AGF strategies toward fragmented subpopulations should utilize pooled genetic diversity from the core population [173] to avoid the diminished genetic diversity of the most divergent or fragmented populations [169,174]. In contrast, AGF toward isolated populations should be avoided [175], except where historic bottlenecks have reduced genetic diversity to the extent of demonstrably reducing fitness and survivability [176,177]. Any advantages of AGF also depend on natural selective pressures toward genetic diversity that favor survival in the wild, irrespective of any loss of genetic diversity [114,117,171,172].

Assisted gene flow toward wild populations can include any life stage; however, various beneficial genetic traits manifest throughout a species’ life history [178], for instance, fungal pathogen lethality varies between the tadpole and the adult stage [179]. Natural selection from early stages favors the retention of endemic genes or beneficial AGF genes along with the loss of detrimental genes [180,181,182,183,184,185]. Therefore, effective AGF strategies toward advantageous genotypes are through egg masses or early larvae. Oocytes could be sourced through on-site collection from wild gravid females, or females in CBPs, and fertilized with genetically diverse sperm [99].

Small, intermittently fragmented wild populations are targeted for AGF programs [186], with the potential benefits and risks being dependent on the population’s most recent fragmentation and any subsequent gene flow. However, the genetics of naturally fragmented populations are challenging to profile and are likely naturally trending toward inbreeding rather than outbreeding [187]. Therefore, AGF could reduce both fragmented and isolated populations’ survival through outbreeding depression and loss of alleles [187,188], the introduction of harmful genes [164,187], influencing male/female incompatibility [189,190], and pathogen transmission [122,123,124,142].

Moreover, the impacts of inbreeding depression and the loss of genetic diversity on the extinction risk of amphibian populations, although widely theorized, are not evidenced in amphibians’ or other taxa’s declines or extinctions through reduced genetic diversity [191], and many species thrive with very low genetic diversity [192,193]. These include island endemic species with effective population sizes of 500–1000 individuals [193], a population size close to the theoretical minimum required to maintain genetic diversity [177]. An increasing number of species of amphibians [10,61] and other taxa [194] are also being repopulated in the wild from a few founders. These cases offer optimism for the survival of small populations of amphibians if a suitable habitat remains for their survival without the potential risks of AGF-supplemented genetic diversity [179,181,195].

Some large-scale repopulation programs are based on the hope that individuals develop unique genotypes that ameliorate or counteract new ecological realities such as lethal exotic pathogens [196]. However, these programs for toads [103,197] and frogs [104] have yet to result in viable populations. Nevertheless, some initially very small natural populations persist despite the prevalence of lethal pathogens [104,179,198]; however, whether this is due to new genotypes, habitat preference, or other behavior at different life stages is uncertain [179,198]. Other programs simply bolster populations through releases, with a recent emphasis on increasing AGF [77].

The potential for outbreeding depression [187] and pathogen transmission [199,200,201] should be evidenced before implementing AGF, and there should be genetic and demographic monitoring both pre- and post-AGF [202,203]. Genetic and demographic monitoring will be a costly process over prolonged periods [177]. A major consideration before implementing costly and potentially risky AGF is that habitat loss may be the overwhelming cause of a species decline. In these cases, habitat protection, amelioration, or provision could maintain species for the time being without the risks and costs of AGF [181,193,195,204], with survival in the wild proving environmental adaptation and genetic fitness [194].

In summary, the benefits of AGF depend mainly on the genetic diversity of the target population, the translocated genotypes, the target population size, proportionate release numbers of individuals with highly beneficial genotypes, the life stage at release, and environmental selection toward the translocated genotypes. The repopulation of captive-bred individuals into the wild can also pose significant risks through outbreeding depression and pathogen transmission. Thorough planning of AGF is essential, including disease risk assessments, genetic management strategies, and post-release monitoring of demographics and population genetics including population viability analysis and computer modeling.

## 3. Reproduction Biotechnologies

Amphibian reproduction modes correlate with those of fishes, and amphibian RBCs have reciprocal practical applications in fish reproductive management, biobanking of germplasm, and CBPs [127,205,206,207,208,209,210]. Here, we provide an overview of amphibian RBCs, including protocols, their application, and future directions, along with the practicalities and ethics of sample collection, and address some recent historical and technical misrepresentations (also see Section 5). Amphibian sperm and oocyte collection, refrigeration of sperm at 4 °C or cryopreserved storage, and artificial fertilization require basic laboratory facilities and animal-handling procedures [211], whereas advanced techniques for cell culture and restoration technologies, such as cloning and assisted evolution, require sophisticated laboratory facilities and technical expertise [82]. Webinars are available online describing details of amphibian reproduction biotechnologies for the hormonal stimulation and collection of sperm and oocytes and their use for in vitro fertilization [212].

### 3.1. Life Stages and Sample Collection

A generalized caution against using early life stages to provide adults for gamete collection was recently published [99]. However, the males of many anuran species mature in less than one year, and both males and females of most species in two years or less [213,214], with salamanders having longer maturation periods than anurans [71,215]. Average oocyte numbers show that many Endangered and Critically Endangered anuran species in the wild can provide surplus oocytes, larvae, and early juveniles [26]. In any case, only a few oocytes are needed for biobanking embryonic stem cells. Different life stages of many Endangered and Critically Endangered species could also be sourced from zoos, CBPs, and private caregiver collections [61,64].

We accessed anuran oocyte numbers in Guirguis et al. [26] taken from IUCN assessments of clutch sizes of 1611 anuran species in total, with 405 threatened species. We added new data from 178 non-threatened anuran species, and then categorized the combined data, using IUCN Red List criteria (Figure 4).

Clutch size declined with increased Red List threatened species status, but not between Endangered and Critically Endangered species, and was highly species-specific. For example, *Atelopus* includes a very high proportion of Endangered and Critically Endangered species [216] and spawns hundreds of oocytes [26]. In contrast, similarly highly threatened *Oophaga* species lay individual eggs [216,217,218]). Both *Atelopus* and *Oophaga* species successfully reproduce in captivity and many colour morphs are held in private caregiver collections (Figure 5, [64]). However, although 96% of *Atelopus* species’ habits are protected, declines and extinctions are continuing [216], and management plans do not emphasize the need for biobanking and disregard the potential of private caregiver collections [217,218]. Nevertheless, a nascent RBC program for *Atelopus* is underway in Ecuador [141].

### 3.2. Vouchering

Vouchering, the process of collecting and preserving physical specimens, can opportunistically provide valuable authenticated sources of biobanked sperm, and other cells or tissues, for RCBs [219,220]. Other conservation benefits of vouchering are taxonomic distinction of metapopulations [126] from genetically divergent isolated populations [204] as distinct management units [221,222,223], informing the epidemiology of amphibian extinctions and declines [196], and providing life history and ecological information required to facilitate CBPs and release programs [142]. Unfortunately, a decline in vouchering has impoverished museum collections that cannot adequately address the looming problem of mass extinctions driven by global heating and other causes [222]. Zoos are making significant contributions to biobanking and vouchering [223], and these contributions could be more widely adopted throughout RBCs, veterinary services, and animal welfare centers.

### 3.3. Reproduction Biotechnologies, Gamete Collection, Donor Stress, and Pathogens

The least stressful reproduction biotechnology is the simulation of natural environmental cues to promote mating and spawning in anurans [71,224] and salamanders [71,215,224]. Cues of temperature and humidity are generally intermittently circannual in terrestrial temperate anurans and continuous in terrestrial salamanders and tropical anurans [225,226], with precipitation generally triggering reproduction [227]. Environmental cues are used by private caregivers for most species to produce large numbers of progeny [64] and in augmentation or repopulation programs [215,224]. Temperature-regulated brumation can also promote gonad maturation, mating, and spawning responses to hormonal stimulation [74,224,228], with males and females of at least one salamander requiring different temperatures to optimize reproductive maturity [229]. Sex differences in response to environmental and social breeding cues are found in one anuran [230].

Amphibian progeny can also be produced through in vitro fertilization with fresh sperm or sperm refrigerated for days to weeks or cryopreserved (Section 3.4, Section 3.5 and Section 3.6). Sperm from the testicular tissues of euthanized males has contributed to foundational [76,93,94,95,231] and recent amphibian RBCs [86,189,232,233,234] and produces the high sperm numbers needed for the efficient production of large numbers of individuals for repopulation, augmentation, and gene-flow programs.

The IUCN, ACAP 2024, mandates that “*In cases where gamete recovery is part of a conservation strategy euthanasia is not recommended*” [61] (p. 294). However, sperm collection from testicular tissue is the preferred method if males are available [71,74,76,86,95,96,231,232,233], as it only requires euthanasia through injection and no further live handling [86,95,96,232,233]. With injection and euthanasia, the dissection of testes and the production of sperm suspensions are achieved in a few minutes [189]. Sperm collection from testicular tissue is preferable (1) when males can be taken from the wild without threatening populations, (2) with surplus males from CBPs [76,86,232,233], (3) with vouchered males (Section 3.1), (4) for the collection of large, high-quality sperm yields from small and especially very small amphibians [71,105], (5) with mortalities after strict pathogen screening [200,201], (6) for the testing of multiple female/male compatibilities [189], and (7) to produce quantities of concentrated sperm suspensions of consistent pH and osmolarity [189]. Sperm collection from testicular tissue is essential for species recalcitrant to hormonal stimulation [10,235].

Hormonal stimulation (misnamed hormone therapy in some amphibian RBC literature [236]) generally requires injection, as with collection from testicular tissue, but does not require euthanasia. Hormonal stimulation has been published for more than 40 species and applied for propagation for release in some species [61,75]. Unfortunately, many studies present sperm yields as concentrations but not sperm numbers, making it difficult to assess the utility of this technique [61,75]. Nevertheless, the utility of hormonal stimulation for many species is limited by generally low concentrations of inferior-quality sperm [10,235]. For instance, hormonal stimulation produced extremely low sperm concentrations and poor-quality sperm with the Endangered frog *Leiopelma hamiltoni* in the basal Anura family Leiopelmatidae, (Figure 6, [235]). A similar response is also likely with *Le. archeyi*, the highest-priority amphibian species on the Zoological Society of London’s Evolutionary Distinct and Globally Endangered (EDGE) list [206]. In contrast, exceptionally large quantities of hormonally stimulated giant salamander semen are readily stripped for *Andrias* (giant salamander) aquaculture in the People’s Republic of China, and in the USA for the development of RBCs for *Cryptobranchus alleganiensis* [71,215,237]. Nevertheless, even if not generally practicable for large-scale progeny production, hormonal stimulation can yield sperm quantities, whether motile or immotile, that are suitable for intracytoplasmic sperm injection (ICSI) [238,239,240,241,242], and low sperm numbers can produce larvae that are then raised to reproductive maturity to produce large numbers of progeny for release [3].

After hormonal stimulation, some anuran species will spontaneously express spermous urine after abdominal massage [243,244,245], or, to sample spermous urine, a cannula or a pipette tip can be inserted into the urethra [10,246]. Salamander sperm can be collected after hormonal stimulation in pre-spermatophores or in gelatinous fluid [247,248], spermous urine [247,248], semen collected by massaging the lower oviduct and cloaca [215,247,249,250], or through the natural deposition of spermatophores [76]. However, hormonal stimulation using injection is stressful to small, delicate salamanders [249], including the highly threatened plethodontids that comprise 65% of salamander species [71,249]. Alternative methods for hormonal stimulation of spermiation in small, delicate salamanders or anurans include topical or nasal application [71,251,252]. The more robust body shape and, consequently, lower skin-surface to body-mass ratio could make small anurans less amenable to the topical or nasal application of hormones than salamanders of the same body weight. On the other hand, anurans have the advantage of a patch of skin in the ventral pelvic region that is permeable to aqueous solutions of hormones [253]. Spermiation was stimulated using a novel technique of hormonally injected crickets fed to salamanders, termed non-invasive oral bioencapsulation. This technique avoids any stress of hormone stimulation for both large and small salamanders [254].

Females of both anurans and salamanders are generally more recalcitrant to hormonal stimulation for oocyte collection than males for spermiation and often require priming doses [74,243,247,255,256,257]. Mature oocytes can then be collected through ovarian excision [95], abdominal massage, or cannulation [71,72,73,74,247,258,259]. Improved targeting of hormonal stimulation has been achieved through ultrasound by identifying mature ovaries in both anurans [260] and salamanders [244,247]. Ovarian excision and spawning into physiological saline (Section 3.6) are low-stress techniques for the sampling of large numbers of oocytes [243,257]. Unseasonal hormonal stimulation of oocytes did not affect oocyte quality in one cool-temperate species [261].

An important factor in selecting gamete collection methods is pathogen transmission [200,201]. Testicular sperm may be contaminated by pathogens such as internal parasites and viruses [157,159], whereas hormonally stimulated sperm in various forms may be additionally contaminated by bacteria, fungi, and external parasites [237].

### 3.4. Sperm Motility and Integrity

Comparative evaluation of spermatological techniques in methods and terminology are confounded by inconsistencies in the evaluation of metrics including motility, membrane integrity, and DNA integrity [75,76,150,205,250,262,263,264,265,266,267]. Basic metrics of sperm motility are the percentages of immotile sperm, activated sperm as non-swimming motion, swimming sperm, and sperm speed or velocity. Sperm motility in amphibians is activated by a lowering of osmolarity from that found in plasma, and higher osmolarities than in nature to extend the period of motility are frequently found in RBC research ([76,234], Section 3.8). The period of motility of anuran sperm is also important during in vitro fertilization and increases as osmolarity rises above that of pond water until the activation osmolarity is reached, although the percentage swimming and velocity of sperm declines at higher osmolarities [76]. Besides various physiological salines where sperm motility is dependent on osmolarity, compounds to stimulate or extend motility such as ATP/adenosine monophosphate [76] or phosphodiesterase inhibitors [234] have failed. Capacitation of anuran sperm and that of salamanders occurs in the oocyte gel [268] and has not been recorded for caecilians [66].

Sperm motility is subjectively assessed by eye [95]; however, meaningful comparative studies need specific setups and methodologies of computer-aided sperm assessment (CASA) to give exact percentages and speeds of swimming sperm [269], along with other motility vectors for more precise analysis [76,270]. Damage to sperm morphology assessed by observation or vital stains includes the integrity of acrosomes [186] and mitochondrial collars or sheaths [266,271,272] (misnamed in some of the amphibian RBC literature as “mitochondrial vesicles” [271,273]). DNA integrity is assessed through comet assays and other tests [228,264,265], with other biochemical tests sometimes used for amphibians [262].

However, although sperm from testicular tissues reliably achieves fertilization, the rate compared to sperm from spermous urine can vary in proportionality from high to low and may depend on reproductive maturity and be species-specific, and little difference was shown between the cryoresistance of sperm from spermous urine or sperm from testicular tissues ([274], Section 3.8). These factors should be considered in comparisons of sperm motility and quality between sperm from spermous urine and from testicular tissue, and further research should be undertaken to provide more meaningful comparisons (Section 3.8).

### 3.5. Refrigerated Storage of Sperm

Refrigerated storage at 0–4 °C is a useful technique for the short-term storage of amphibian sperm. Refrigerated storage enables delayed in vitro fertilization when there is asynchrony between sperm collection and oocyte availability [262] or for transport between breeding groups [99]. Refrigerated storage for ~10 min can acclimate sperm to cryoprotectants [266] and be used to delay cryopreservation for days to weeks [76,96]. Refrigerated storage of anuran sperm is highly successful in cadavers [228,245], whole testes [96], macerated testicular tissue [96], and spermous urine [75]. However, refrigerated salamander sperm only remains motile for a day to several days [71,76] or for weeks when held in spermatophores [275]. Oxygenation [276,277] and antibiotics to reduce bacterial concentrations improve sperm storage periods while maintaining sperm concentrations, with antibiotics reducing the chance of pathogen dissemination [75,276,277,278,279,280]. Refrigerated sperm may be inactivated in hypertonic solutions, in testes, or in testicular macerates, with motility minimized at low temperatures in spermous urine [245,278,280,281,282].

### 3.6. Sperm Cryopreservation and Freeze Drying

Cryopreservation is a freezing process that protects sperm using special solutions to create sperm cryosuspension and tailored freezing rates. The cryopreserved sperm can then be held indefinitely in biobanks [75,76,283]. Sperm cryopreservation uses aqueous formulations, termed cryodiluents, mixed with sperm from testicular tissues or spermous urine to produce cryosuspensions that are then frozen [75,76]. Cryodiluents are formulated from sperm- penetrating cryoprotectants, such as dimethyl sulfoxide (DMSO, [93,94] or dimethylformamide (DMFA, [228,266]), and non-penetrating saccharides or salts, with supplements such as buffers, fetal bovine serum, and antibiotics [75,76]. Fertilization was first achieved with cryopreserved anuran testicular sperm in 1996 using DMSO [93] and with hormonally stimulated sperm in 2011 using DMFA, then a novel cryoprotectant to amphibians [266]). Mouse sperm has also been freeze-dried and then stored at room temperature [284] and remained viable, but this technique has not been tested in amphibians, either for fertilization [71,76] or for ICSI [238,239,240,241]

A broad canvas of the RBC community in 2019 recommended 5–10% (*v*/*v*) DMSO or DMFA and 1–10% (*w*/*v*) saccharide as cryoprotectants for amphibian sperm and the use of slow to moderate cooling rates [76]. Nevertheless, ranges of 12–15% DMSO have proved successful with anurans [93,94,96,228,232,233,266], with 12% DMFA proving superior to DMSO in a pioneering study with urinal sperm [266] and then generally used in other studies of hormonally induced sperm [270,274,283]. Both fast and slow cooling rates have recently been successful [75,76,270]. Cryosuspension osmolarity has shown a significant effect on cryopreservation in some studies [232], while other studies have shown little effect [274]. The freezing of sperm over a range of freezing rates can be achieved by constructing a simple and inexpensive device first used for fish sperm [285], with cooling rate impacts on amphibian sperm found in [270]. The broad differences in successful cryopreservation protocols, even within the same species, may be due to subtle variations in techniques for cryoprotectant penetration, freezing regimes, the thawing and washing of cryoprotectants from cells [75,76], seasonal effects on sperm quality [281], in vitro fertilization techniques lacking meaningful baselines (see Section 3.8), unconcise sperm-quality metrics (Section 3.4), and sperm genotypic and phenotypic characteristics [125,283]. There was little difference between the cryoresistance of urinal sperm or sperm from testicular tissues in *Bufo bufo* [274].

### 3.7. Oocyte Storage

The storage of oocytes enables delayed in vitro fertilization if there is asynchrony between the availability of oocytes and sperm [75,76,286]. Because of their high yolk content and large size, the cryopreservation of anuran oocytes is not yet practicable [287]. The viability period of unfrozen oocytes during storage depends on a species’ natural spawning temperature, the storage temperature, and the oocyte’s osmotic, ionic, and gaseous environment [74,96,231,282,288,289]. Early conventions for oocyte storage assumed that ionic formulations slowed oocyte gel hydration (described as hardening), which blocked sperm penetration [158,288,289]; however, research over a broader range of species showed that the hydrated oocytes of some species are fertilizable for an hour or more [231]. Unhydrated and refrigerated ovarian oocytes and post-ovarian oocytes of cool-temperate species have remained viable for many days [282], whereas a pressurized gaseous environment extended storage life further [290].

Oocyte storage is highly dependent on the target species’ natural spawning temperature, where the oocytes of species spawning in cold water may remain viable for many days [282], and tropical or subtropical species only for hours [96,288,289]. Studies with a single species showed that a percentage of post-ovarian oocytes undergo a very rapid loss of fertility [189], possibly a female sperm choice mechanism for greater progeny heterozygosity with polyandrous species [291], or to maintain sub-population environmental adaptability where female/male genetic incompatibility lowers fertilization rates in anurans with little dispersal ability [189,190]. In other species, some oocytes resist fertilization during spawning, leaving fertilizable oocytes for later fertilization [151]. However, the physiological or genetic mechanisms behind selective oocyte fertility loss over time, and any subsequent genetic biases toward progeny, are undetermined. Therefore, besides species-specific temperature effects, the mechanisms for the loss of oocyte viability in aqueous solutions generally appear to be through the diffusion of ions or proteins from the oocyte gel needed for sperm motility or for oocyte metabolism [292,293,294,295] rather than gel hydration [288,289]. Specific genotypic mechanisms also influence the storage period and even extend to individual oocytes [189].

The cryopreservation of amphibian oocytes or early embryos has no parallel with techniques used for mammalian oocytes, which are very small, ~0.08–0.20 mm in diameter, and have minimal yolk content [296]. However, the cryopreservation of amphibian oocytes or embryos includes many parallels with fishes [207,208,287] due to both possessing highly fatty and structured egg-yolk and oocyte sizes. Nevertheless, the success of cryopreservation techniques for fish embryos up to 0.8 mm in diameter Ref. [207] is challenging to apply to the large size of more than 95% of amphibian oocytes that are over 0.9 mm in diameter (Figure 7). Therefore, currently, the most promising techniques to perpetuate the amphibian female genome is using cryopreserved biomaterial for heterocytoplasmic cloning (Table 3, ibid) or stem cells to generate ovaries in surrogate species (Section 5, [287]).

### 3.8. Fertilization; Sperm Concentrations, Fertilization Periods, and Rates

The critical endpoint of fertilization rates when comparing sperm concentrations and quality is confounded by varying techniques and terminologies, even with respect to misinformation in historic attributions. The term “fertilization” is generic and includes in vitro (traditionally also “artificial fertilization”) fertilization, where sperm is placed over oocytes, as presented in [189,286,308], but also artificial insemination with sperm placed internally [309] and intracytoplasmic sperm injection (ICSI) with sperm placed into an oocyte [238,239,240,241]. In vitro fertilization is the critical endpoint of most amphibian BBC research and has been achieved with scores of anuran species and tens of salamander species [61], but artificial insemination has been trialed with only one salamander species [309].

The first book devoted to amphibian BBCs published in 2022, “*Historical Perspectives on the Development of Amphibian Reproductive Technologies for Conservation*” ([286], p. 1), stated that in vitro fertilization in amphibian RBCs was “dry fertilization” and attributed “dry fertilization” as a pioneering experimental achievement to Rugh, 1961, in the USA [310]. Historically, dry fertilization was pioneered in 1856 in Russia, where inactivated fish sperm was mixed with oocytes, and then, after a short period, the sperm was activated with water to achieve fertilization [311]. In contrast, Rugh first published the placement of activated sperm on anuran oocytes in 1934 [312]. This technique was then furthered in other early studies [288,289], first used in amphibian RBCs in Russia in 1996 [94], and subsequently generalized in amphibian RBCs [228,243,244,245,257].

Our knowledge of artificially collected sperms role in in vitro fertilization is limited by the literature lacking comparable sperm-quality metrics (Section 3.4 [313,314]), sperm concentrations and fertilization periods [95,231,266,314,315], sperm activation and fertility in different osmotic environments [75,76,234,250,315,316], and species- and maturation-specific differences between sperm from testicular tissue and from spermous urine ([264], Section 3.4). The effect of osmolarity can even extend to the osmotic environment of donor males [316]. Fertilization curves are the endpoint of sperm quality; however, few publications have included amphibian fertilization curves, which were first presented in the general scientific literature in 1975 [314] and the RBC literature in 1998 [96]. Furthermore, the fertilization period in most amphibian studies is 10 min and refers to concentrated sperm suspensions at high osmolarities [96]. However, comprehensive fertilization curves at different sperm concentrations show that fertilization rates with *Rhinella* (*Bufo*) *arenarum* increased for at least 30 min post application, with the post-application period being more influential than linear sperm concentrations on fertilization rates (Figure 8, [314]).

Other fertilization curves show saturated fertilization with sperm from testicular tissue at concentrations of 2.5 × 10^5^ mL^−1^ after 60 min [315], ~10^4^ mL^−1^ after 60 min [231], ~10^6^ mL^−1^ after 10 min [96]), and 10^5–7^ mL^−1^ after 60 min [289]. The fertilization rate, in general, will also depend on the relatively short motility period and lower quality of cryopreserved sperm compared with fresh sperm [317,318]. For instance, after 15 min application, *Rana temporaria* spermous urine at ~10^6^ mL^−1^ provided ~45% fertilization, while testicular sperm provided only ~2% fertilization. Spermous urine achieved saturated fertilization at ~10^7^ mL^−1^ but testicular sperm only at ~10^8^ mL^−1^ [266].

The need for robust fertilization curves to compare fertilization rates was exemplified through much higher fertilization rates than expected with cryopreserved sperm from the spermous urine of *Anaxyrus* (*Bufo*) *fowleri*. Control fresh swimming sperm provided ~85% saturated fertilization at ~4.2 × 10^6^ mL^−1^ concentration, whereas after cryopreservation, swimming sperm at only ~2% concentration of the control provided ~20% fertilization [319], even though the fertility of cryopreserved sperm is lower than fresh sperm [266,274]. A robust fertilization curve could explain this apparent enigma and, in general, provide baselines for meaningful comparisons among all studies.

## 4. Advanced Reproduction Biotechnologies (aARBs)

Advanced Reproduction Biotechnologies (aARBs) can perpetuate species through the cryopreservation of cultured or uncultured somatic cells [320] with subsequent cloning [99,208,287,296,321,322], or stem cells used for the generation of gametes in surrogate species [77,80,81,287]. The generation of gametes includes the implantation of cryopreserved immature ovarian follicles, primordial germ cells, or induced pluripotent stem cells to generate chimeras [73,77,287]. These techniques were first introduced to the amphibian RBC literature in 1999 [70].

### 4.1. Cloning

Heterocytoplasmic cloning enables species restoration solely from biobanked nuclei through surrogate species [99,208,287,296,321,322], and the critical need to develop heterocytoplasmic cloning has not received the attention it deserves in amphibian RBCs. To perpetuate amphibian biodiversity, heterocytoplasmic clones must develop to reproductively mature and fertile females to produce oocytes for fertilization with cryopreserved sperm. Homocytoplasmic anuran clones from embryonic cells were developed to late blastula as early as 1952 [297], with heterocytoplasmic cloning first accomplished with anurans in Japan and with salamanders in France (Table 3, ibid). Heterocytoplasmic clones were then developed to the gastrula stage in 1957 [298], early neurula in 1958 [299], adults in 1961 [300], with naturally mating and spawning adults producing viable offspring in 1963 [97,98,301], and second and third generations from 1963 to 1972 [302,303], and from as early as 1971/1972, adult heterocytoplasmic clones were produced from a wide range of other anurans [304,305] and also from salamanders (*Pleurodeles* sp.) (Table 3, ibid; [306,307]. In 1998, nuclei from cryopreserved totipotent cells were developed to the gastrula stage [322] using the techniques of [321]. These foundational studies and the extraordinary rate of the development of de-extinction (species restoration) projects for other taxa relying on cloning [323] herald a new age in amphibian cloning that will lead to species perpetuation solely from cryopreserved nuclei. This will greatly reduce the need for CBPs for species unable to survive in the wild or CBPs for species of little cultural interest. This advance will not only relieve some of the financial burden through maintaining live amphibians but will also have considerable animal welfare benefits.

The first book devoted to amphibian RBCs published in 2022, “*Historical Perspectives on the Development of Amphibian Reproductive Technologies for Conservation*” ([286], p. 1), attributed “*one of the most noteworthy and groundbreaking accomplishments toward amphibian reproductive technologies*” to Gurdon’s 1962 publication of amphibian homocytoplasmic cloning in the USA [324]; this accomplishment also resulted Gurdon receiving a Noble Prize in 2012 [325]. However, Gurdon’s homocytoplasmic cloning was not pioneering and extended Briggs and King’s 1962 homocytoplasmic cloning of amphibians in 1952 [297], and neither concerned heterocytoplasmic cloning a technique essential for amphibian species perpetuation [286]. Furthermore, the 2012 Nobel Prize for Physiology or Medicine was awarded jointly between John Gurdon and Shinya Yamanaka [325], not for cloning, but for the reprogramming of nuclei to pluripotency [325], as discussed by Gurdon in [326] his acceptance speech “The Egg and the Nucleus: A Battle for Supremacy” [327]. Reprogramming of nuclei to pluripotency was not used in the heterocytoplasmic cloning listed in Table 3, as pluripotent nuclei are readily available from early amphibian embryos (Table 3, ibid).

### 4.2. Assisted Evolution

Traditional methods of assisted evolution include the selection of desirable traits in domestic species [328], as found in amphibians with private caregivers’ colored morphs [64]. Other assisted-evolution techniques for amphibians include the natural selection of beneficial traits through mass releases [103,104] and approaches toward genetic engineering to provide pathogen immunity or adaptability to the climate crisis [61,129,329,330]. In contrast to most amphibians, the benefits of assisted evolution in mammals are challenged by their low reproductive rates [82]. Nevertheless, assisted evolution is a crucial strategy to select or increase amphibian genetic diversity or for species restoration [3,61,77].

## 5. Ethics and Communication

By addressing ethical principles, communication challenges, and cultural normalization, we can fully realize the potential of RBCs [3,44,45,46]. The ultimate standard for the ethical treatment of all animals is “*The physical and psychological well-being of an animal* (sic, in captivity)*. It is good or high if the individual is fit*, *healthy*, *free to express natural behavior*, *free from suffering*, *and in a state of wellbeing*.” [331].

Ethical principles dictate that RBC practitioners are moral agents responsible for the well-being of their moral subjects, the species of concern [332,333]. Practitioners’ ethical motivation toward amphibian RBCs is driven by a responsibility to perpetuate amphibian biodiversity while avoiding unnecessary harm [61]. Practitioners are guided by the precept, “*How would I like it if I were them?*”, an ethical principal present in all cultures with ethical traditions [333]. RBC animal-ethics standards include amphibians possessing sentience through the cognition of stimuli without association or interpretation [333,334,335], a standard that can even extend to invertebrates [336].

In practice, the ethical use of RBCs depends on a balance between (1) the sentience of the target individuals [332,337], (2) the species’ role as a biospheric entity, thereby benefiting other moral subjects [338]; (3) support for intergenerational justice toward the environment [44,45,46]; and (4) the principle of the greatest good for the greatest number as applied to the species perpetuation [339]. Ethical considerations depend on the justification for conducting the technique, its efficiency, general applicability, and associated stress to the targeted individuals (Table 4, ibid). Once an RBC protocol or program using animals is justified, the three basic animal-ethics principles of refinement, reduction, and replacement in research or application can be applied [340,341,342]. More research on stress during confinement and handling is needed to inform ethical standards of amphibian RBCs [163].

Refinement can be achieved by preferentially using environmental simulation to promote mating and spawning in CBPs [211,331], and through procedures that minimize stress and pain during the collection of sperm and oocytes (Section 3.2, [254,343]). Reduction can be achieved through reducing the number of individuals in CBPs by using stored sperm and by avoiding extensive hormonal stimulation trials to optimize hormonal sperm collection when large yields of sperm could be reliably collected from testicular tissues (Section 3.2). Replacement can be achieved by programs maintaining genetic diversity through biobanking of germplasms, somatic cells, or tissues for use in CBPs and in supplementation, repopulation, or assisted gene-flow programs (Section 3.2, [3,75,76]), and through optimization of these activities through genetic and demographic modeling (Section 2.2).

Biotechnologies for sperm collection using the relatively stressful technique of hormonal stimulation, when compared to collection from testicular tissue, have been subject to extensive research involving more than 40 species [61,343]. However, hormonal stimulation requires the handling of individuals, and cannulation over extended periods generally only yields low sperm numbers and concentrations [235]. Furthermore, dose-response curves for two hormones are recommended to optimize hormonal stimulation before its general use for a species [10], which requires at least 32 experimental individuals. However, if mature males are available, high sperm yields can be collected through the reliable low-stress technique of testicular tissue to provide the sperm numbers for the fertilization of large numbers of oocytes (Section 3.2). In either case, even a few biobanked sperm can provide genotypic variation or produce mature adults to provide large numbers of progeny.

Despite these ethical limitations, the collection of spermous urine through cannulation is critical and justified to provide sperm known to be equivalent to those expressed in natural spawning for the basic science of spermatology and its contribution to amphibian RBCs [266]. Spermatology is dependent on knowledge of sperm structure, physiology, and motility mechanisms, particularly with respect to viscosities found in internally fertilizing amphibians and the physical structure of oocyte gel and female sperm-choice mechanisms [71]. Institutional animal-ethics requirements should be sympathetic and flexible toward the critical need for further research to develop amphibian spermatology and for the development of RBCs to perpetuate amphibian biodiversity.

Besides animal welfare, the ethics of amphibian RBCs extend to human/animal interactions through emotional and social engagement and the broad ecological impacts of repopulations or translocations [3,81,82,334,335,344]. Furthermore, speciesism, where a preference for biobanking a species is based on political, institutional, or charismatic criteria rather than on conservation or phylogenetic status requires ethical scrutiny [206]. There is considerable engagement of the theological community in the need to support biodiversity conservation [345], and avoiding language that may be theologically misinterpreted, such as “resurrection” [85] for species restoration and “playing God”, helps to ensure broader public support [85,346,347,348]. Theologies can also contribute to biospheric sustainability through social and ethical models that respect nature [349]. Although subject to ethical scrutiny with respect to speciesism, the popularity of species restoration is shown by the global reach of the Colossus de-extinction project [307], and ethical considerations include its potential for biospheric sustainability, profound community outreach, and its financial and scientific support of the field conservation of threatened species [3,344].

## 6. Current and Future Applications

The development of amphibian RBCs has progressed to the collection and storage of anuran and salamander sperm and oocytes and their use for in vitro fertilization to produce sexually mature adults [71,86,87,233]. These RBCs can support either institutional or private CBPs, along with repopulation or translocation programs [3] and assisted gene flow (Section 2.2) However, costly programs for wild populations should be highly targeted toward ecologically, phylogenetically, or culturally significant species that are likely to eventually independently survive in the wild for at least decades [61,126,206]. This targeting is particularly relevant in consideration of the rapid and accelerating anthropogenic modification of the biosphere (See Section 1).

Species not expected to survive in the wild, irrespective of the environmental targets of COP 15, 16 [2,3], or COP 28 [3] should be subject to CBPs supported by biobanked sperm or perpetuated solely in biobanks (see Section 3). Only through addressing these issues will the amphibian RBC community satisfy their ethical responsibilities toward providing reliable and cost-effective intergenerational justice. Reproduction has been well studied in only about 5% of caecilian species, and we must first learn more about their basic process of reproduction and its controls before we can generalize artificial augmentations [350]. Therefore, caecilian species offer exceptional challenges and opportunities for the development and application of RBCs because of many species’ internal fertilization and development and the maternal care of juveniles in all terrestrial species [71,89,90,91].

The high costs of implementing recovery-based conservation are attributed to maintaining or providing specialized habitats, extensive research and monitoring, conservation breeding and reintroduction programs, and the need for professional management [3,351]. The financial burden of meeting these global biodiversity conservation targets by reducing extinction risk through interventions along with establishing and maintaining protected areas is substantial, with estimates of billions a year [352], with mammals and birds typically requiring the most resources [353]. To address these challenges, a globally inclusive approach is needed, focusing on developing highly cost-efficient RBC facilities in regions with high amphibian diversity that welcome community engagement and international collaborations [3,354].

The short-term prevention of the extinction of amphibian species through RBCs to maintain, repopulate, or translocate amphibian populations in the wild [3] has already been reliably and economically demonstrated. Between 2007 and 2017, there was a ~60% increase in amphibian CBPs, including 80 species, or 2.9% of all species, with half of these CBPs involving repopulation or augmentation and 70% having some success [10,61]. However, these approaches must recognize a future of dramatic ecosystem collapses driven by global heating leading to mass extinctions [13,47,48,49,54]. The establishment of biobanks of amphibian cells and tissues will provide cost-effective and reliable species perpetuation [3,68,77,78,79,80,81,82,83].

Exciting opportunities for amphibian RBC research and development include cloning and somatic-cell techniques for species restoration (de-extinction) [323]), extraterrestrial biotechnologies [3,355,356], and terraforming for colonization [3,355]. Terraforming conceptually includes the past, current, and future anthropogenic modification of Earth’s biosphere as extended to potentially interplanetary and interstellar extraterrestrial ecosystems along with their biodiversity [3,355].

Public outreach for amphibian conservation includes educational programs, community engagement, media campaigns, and interactive experiences [357] to foster empathy, promote citizen science, and encourage sustainable practices [3,61,64,358]. Perceptions of conservation success are generally perceived in terms of species and habitat improvements, effective program management, and the application of science-based conservation including RBCs [61,358].

The 2024 ACAP provides a guide for capturing outcomes, identifying gaps, and measuring progress in conservation efforts [61]. However, the ACAP framework was limited to supporting “*Amphibians thriving in nature*” (in the wild), along with the Amphibian Ark neglecting species that cannot foreseeably be returned to the wild [359], and both generally disregard transformational changes to species management [3]. “Chapter 12, *Amphibian assisted reproductive technologies and biobanking*” of the 2024 ACAP, in the “*Priorities and recommendation”* section, does not mention species perpetuation through biobanking of somatic cells or tissues, cloning, and species restoration [61]. Furthermore, the 2024 ACAP “*Chapter 11. Conservation breeding*” of the 2024 ACAP is not inclusive of private caregivers’ potential to contribute to the perpetuation of amphibian biodiversity [64]. The disregard of many internet sources for the benefits of the biotechnical aspects of amphibian RBCs and biobanking [360] shows the challenges facing the effective advocation and popularization of the full potential of RBCs.

The advocation and popularization of amphibian RBCs include engaging cultural discourses involving assisted evolution [361], cloning and species restoration [82,362], synthetic biology [355,363,364], and theological debates about humanity’s relationship to nature through synthetic biology, “resurrecting” species, “playing God” [361,362,363,364], and utilizing scientifically based terminologies popularized through the public media [3,61,105,106,107,271,365]. Solid RBC career streams will attract ambitious and talented researchers and activists to build RBC projects by successfully competing for influence and resources [3,105]. Initiatives should focus on species perpetuation to ameliorate amphibian mass extinction and be inclusive of the broadest global community [3,345]. Community popularization of advanced reproduction technologies is exemplified by the highly successful Colossal project, where their central theme of de-extinction (species restoration) is extended for their target species to community-based field-conservation projects [323,344], advocating bioengineering as an initiative to help heal the Earth [366,367].

## 7. Conclusions

We have shown that the perpetuation of amphibian biodiversity, rather than focusing on the unachievable goals of many amphibians thriving in the wild, should recognize the community’s responsibility with respect to intergenerational justice to adapt to the environmental and cultural realities of the Anthropocene [368,369], including the sixth mass extinction [6,7,15,47,368] and humanity’s interplanetary and interstellar colonization. This transformation should harness the potential of biobanking of germplasm, somatic cells, or tissues and species restoration to perpetuate species and be inclusive of all global participants [3,64]. A greater focus is also needed to garner global support and international engagement toward developing RBCs in highly biodiverse regions, especially developing countries, and the provision of independent finance. An independent, democratic, and inclusive global organization representing an increasingly assertive multipolar world is needed to advocate these needs [3]. Finally, will humanity, over the coming millennia, look back to this century relegating plausibly thousands of amphibian species that will not continue to exist in the wild to extinction [359,370]? Or, alternatively, will we face the challenges of the Anthropocene presented by inevitable biospheric modification, especially through global heating [48,49], and perpetuate species through realist innovative, cost-effective, and reliable biotechnologies [3] supported by broader community inclusion in a multipolar world [3,53]?

## Figures and Tables

**Figure 1 animals-14-03395-f001:**
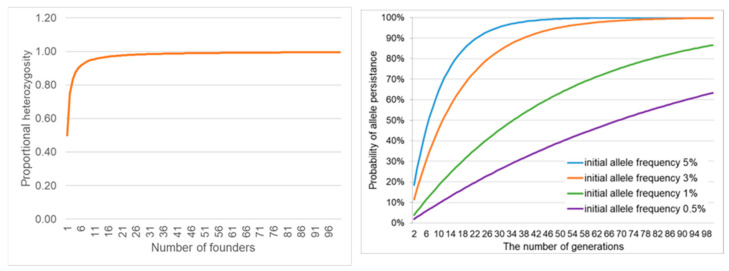
The left panel illustrates the proportional heterozygosity/genetic diversity (*y*-axis) through random sampling of individuals (*x*-axis) for a founding population. The right panel illustrates the probability that an allele will persist in the next generation (*x*-axis) dependent on the number of equally representative reproducing individuals in the captive population (*x*-axis).

**Figure 2 animals-14-03395-f002:**
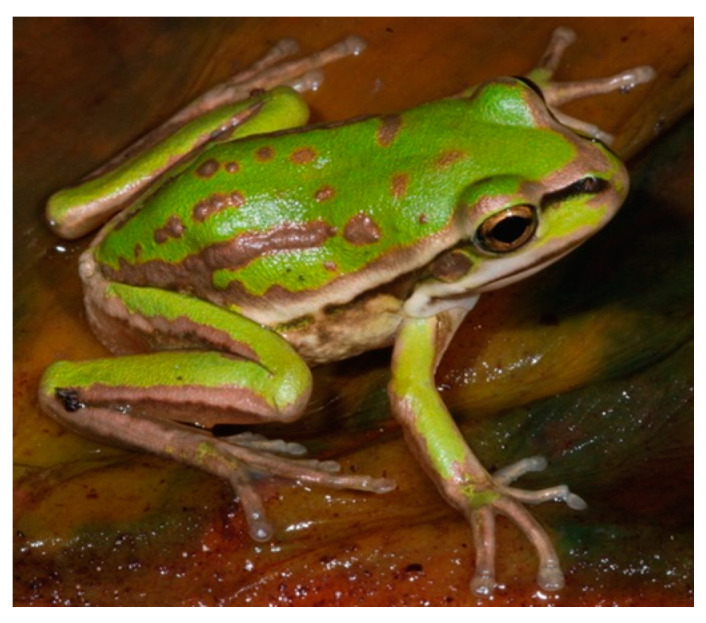
The most common morph of the green and golden bell frog, *Ranoidea (Litoria*) *aurea*, includes both green and gold coloration (image—© 2008 Dr. Peter Janzen).

**Figure 3 animals-14-03395-f003:**
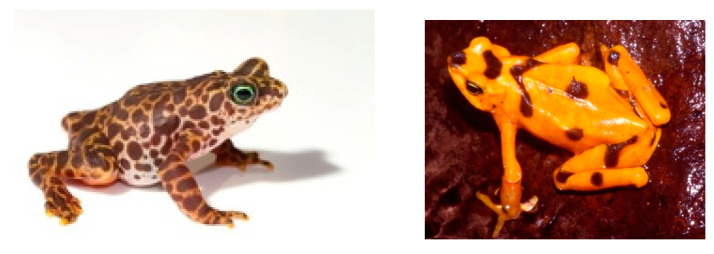
Left, *A. certus* (Image—© 2019 Brian Gratwicke, Creative Commons Attribution 3.0 (CC BY 3.0)); right, *A. zeteki* (Image—© Peter Janzen).

**Figure 4 animals-14-03395-f004:**
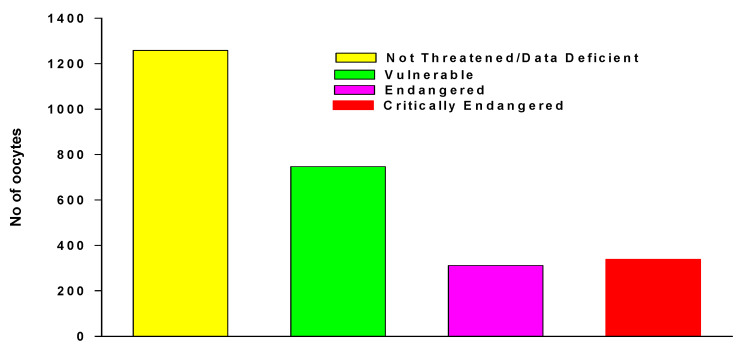
The average anuran clutch size by IUCN Red List endangerment status.

**Figure 5 animals-14-03395-f005:**
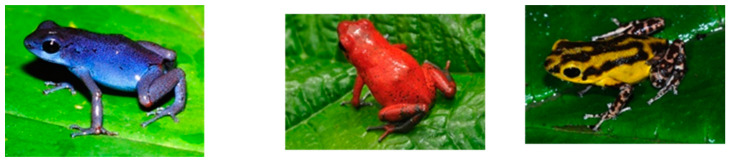
Left, blue morph; middle, red morph; right, yellow morph. Different color morphs of the strawberry poison dart frog, *Oophaga pumilio*, and many other amphibian species are very popular with private caregivers and have a long history of domestication (Images Peter Janzen).

**Figure 6 animals-14-03395-f006:**
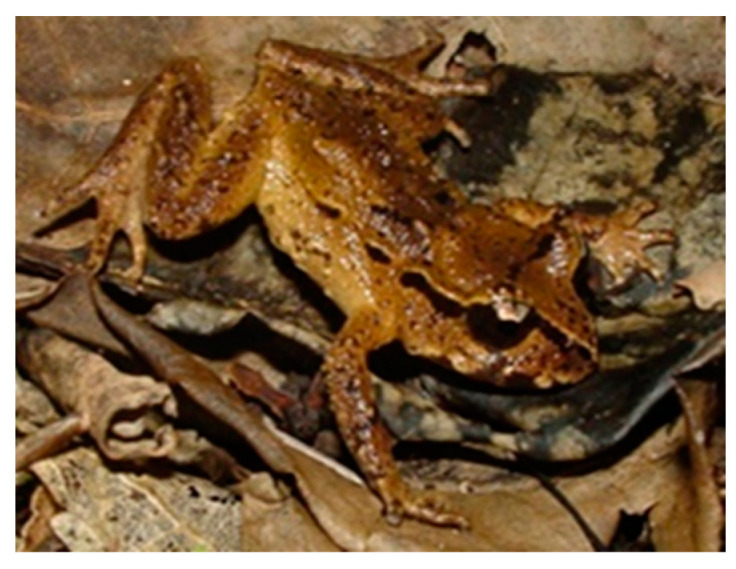
Hamilton’s frog (*Leiopelma hamiltoni*), where hormone stimulation produced extremely low sperm concentrations of poor-quality sperm. Image by Phil Bishop. Attribution ShareAlike 2.5. https://en.wikipedia.org/wiki/Hamilton%27s_frog#/media/File:Leiopelma_hamiltoni02.jpg (accessed 19 September 2024) Phil Bishop, CC BY-SA 2.5 https://creativecommons.org/licenses/by-sa/2.5, (accessed 19 September 2024) Wikimedia Commons.

**Figure 7 animals-14-03395-f007:**
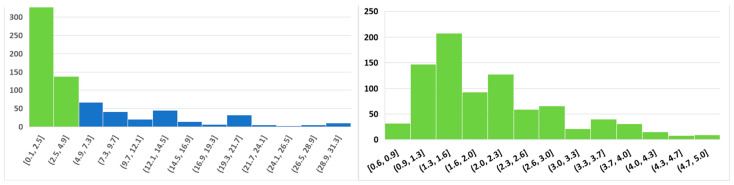
The *x*-axis shows the number of anuran species, and the *y*-axis shows oocyte volumes (mm3). Left panel, 805 species with oocyte diameters up to 4.9 mm and volumes up to 31.3 mm^3^. Right panel, ~430 of the 805 species from the green bars in the left figure with oocyte diameters ≤ 1.8 mm and volumes ≤ 5.0 mm^3^.

**Figure 8 animals-14-03395-f008:**
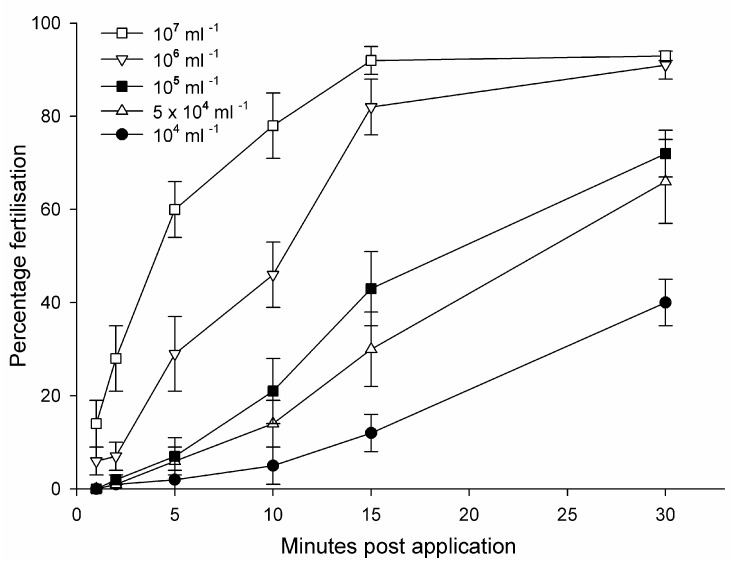
*Rhinella* (*Bufo*) *arenarum* (Bufonidae) fresh testicular sperm concentrations, the percentage of in vitro fertilization, and minutes post application before washing with fresh water [314].

**Table 1 animals-14-03395-t001:** Alternative approaches to the numbers of biobanked founder individuals for maintaining ex situ genetic diversity. M = male, F = female, CBP = conservation breeding program.

Population	M	F	Application	Year	Reference
Metapopulation	5	5	27 threatened species.	2005	[133]
Metapopulation	28	25	CBP	2007	[120]
Metapopulation	25	-	CBP	2012	[134]
Fragmented/sub-pop.	10	-	AGF 4 years, depending on kinship	2022	[135]
Geographic clusters	10	20	Geographic clusters	2022	[130]
Metapopulation	20+	20+	CBP	2024	[118]

**Table 2 animals-14-03395-t002:** The number of individuals to be housed in a CBP to maintain 90% of the original genetic diversity from 25 male and 25 female founders over programs of 25 years, based on age to maturity, longevity, and individual studbook management (Individual) or group management (Group). * = Minimum population sizes of more than 100 are recommended to allow for unexpected loss of individuals [120]. # = Only group management because of short longevity.

Examples of Genera	Age to Maturity	Longevity	Individual	Group
*Acris*	<1 year	1 year	1590	#
*Eleutherodactylus*, *Nectophrynoides*, some *Hyperoliidae*	<1 year	2–5 years	400	400
*Hylidae*, some *Hyperoliidae*, *Scaphiophryne*	1–5 years	<5 years	135	265
*Dendrobatidae*, *Typhlonectes*, *Tylototriton/Echinotriton*, *Theloderma*, *Cynops*, *Leptodactylus*, *Ceratobatrachus*, *Mantella*, *Atelopus*	1–5 years	5–15 years	70 *	140
*Salamandra*, some *Ambystoma*	1–5 years	>15 years	60 *	80 *
*Cryptobranchus*, *Andrias*	>5 years	>15 years	45 *	80 *

**Table 3 animals-14-03395-t003:** Heterocytoplasmic somatic cell nuclear transfer (SCNT) cloning in amphibians.

Nucleus Donors	Recipients	Results	Year	Ref
*Rana pipiens*	*Aquarana* (*Rana*) *catesbeiana*	Died late blastula	1952	[297]
*R. n. brevipoda*	*R.n. nigromaculata*	Metamorphs	1957	[298]
*R. pipiens*	*R. sylvatica*	Late blastula/early neurula	1958	[299]
*R. n. nigromaculata*	*R. n. brevipoda*	Adults	1961	[300]
*R. pipiens*	*R. palustris*	Post-neurula	1963	[301]
*R. n nigromaculata*	*R. n. brevipoda*	Adults–F1 reproduction	1963	[97,98]
*R. japonica*	*R. ornativentris*	F2	1963	[302]
*R. nigromaculata* *R. temporaria*	*R. brevipoda* *R. japonica*	F3	1972	[303]
*R. japonica* *R. temporaria*	*R. temporaria* *R. japonica*	Adults	1972	[304]
*R. brevipoda* *R. plancyi* *R. brevipoda* *R. esculenta*	*R. plancyi* *R. brevipoda* *R. esculenta* *R. brevipoda*	Adults	1972	[305]
*Pleurodeles waltlii* *P. poireti*	*P. poireti* *P. waltlii*	Adults	1971	[306]
*P. waltlii* *P. poireti*	*P. poireti* *P. waltlii*	Adults	1972	[307]

**Table 4 animals-14-03395-t004:** Ethical considerations with techniques for natural or induced mating and spawning and the collection of sperm or oocytes. Note: there has been little research on stress through the handling of amphibians. Consequently, our estimates of stress are simply relative to the alternative techniques. However, even confinement in boxes alone, without regrading other handling, as used for hormonal stimulation, causes measurable stress [163].

Technique	Injection	Application Stress	Collection Technique	Collection Stress	Yield	Donor Size Limitation	References
Mating and spawning							
Natural mating and spawning	None	None	Fertilized eggs	None	Very high	None	[214,215]
Hormonally induced mating and spawning	Yes	Moderate	Fertilized eggs	None	Very high	None	[74,224]
Oocyte collection							
Hormonal stimulation oocytes with spawning into solutions	Yes	Moderate	Spawning into physiological saline	Low	High to very high	None	[243,257]
Hormonally stimulated ovarian oocytes	Yes	Moderate	Abdominal massage or cannulation	High	Low	Avoid for small species	[74,243,247,255,256,257]
Excision of hormonally stimulated ovarian oocytes	Yes	Moderate	Direct sampling	None	Very high	None	[243,257].
Sperm collection							
Collection from testicular tissue	Yes	Moderate	Macerating testicular tissue	None	Very High	None	[71,74,76,86,95,96,231,232,233]
Hormonal stimulation sperm	Yes	Moderate	Urination	Moderate	High to none	Avoid for small species	[243,257]
Hormonal stimulation sperm	Yes	Moderate	Cannulation	High	Moderate—very low	Avoid for small species	[61,75]
Hormonal stimulation sperm—topical, nasal, oral, or food item	None	Low	Cannulation	High	Moderate—very low	Possibly for small species	[71,251,252]

## Data Availability

Data concerning oocyte numbers and sizes are currently being used to support several research articles on the evolutionary ecology of amphibian sperm. Nevertheless, the data presented in this study are available, in confidence, on request to the corresponding author.

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
