# Peer review of "The Sixth Mass Extinction and Amphibian Species Sustainability Through Reproduction and Advanced Biotechnologies, Biobanking of Germplasm and Somatic Cells, and Conservation Breeding Programs (RBCs)"

_animals, 2024, doi:10.3390/ani14233395_

Round 1
Reviewer 1 Report
Comments and Suggestions for Authors
The authors discuss how reproductive biotechnologies, biobanks, and conservation breeding programs (CBRs) enable species restoration and offer a critical transformative change that can economically and reliably perpetuate species regardless of environmental goals.
However, we feel that this review could be more focused on reproductive techniques such as biobanks and conservation breeding programs than on genetic management and political and cultural engagement and future applications. The review is too long and should be shortened and more concise.
Author Response
Reviewer 1. Report
The authors discuss how reproductive biotechnologies, biobanks, and conservation breeding programs (CBRs) enable species restoration and offer a critical transformative change that can economically and reliably perpetuate species regardless of environmental goals.
However, we feel that this review could be more focused on reproductive techniques such as biobanks and conservation breeding programs than on genetic management and political and cultural engagement and future applications. The review is too long and should be shortened and more concise.
Response: The object of the review is to further extend the field of amphibian reproduction biotechnologies into genetic management and political and cultural engagement and future applications. This extension is very important as most previous publications have not adequately addressed these topics of critical importance to species perpetuation. The reproduction technologies section is simply to contextualise these topics, and to provide a background for those unfamiliar with the numerous research publications and reviews on this subject including a recent book, and webinars devoted to the topic as referred to in the review.
Reviewer 2 Report
Comments and Suggestions for Authors
1. Brief Summary
The manuscript aims to address the impending sixth mass extinction and the role of reproductive biotechnologies, biobanking, and conservation breeding programs (RBCs) in sustaining amphibian species. It highlights the transformative potential of these technologies in preserving biodiversity despite environmental challenges. The paper's main contributions lie in providing a comprehensive overview of RBCs, discussing future applications, and addressing ethical, philosophical, and geopolitical issues. The strength of the manuscript is its ambitious scope and its attempt to integrate diverse aspects of species conservation into a cohesive narrative.
2. General Concept Comments
This reviewer commends the authors for their efforts in compiling an important collection of hundreds of pieces of information on assisted reproduction in amphibians. The comments and criticisms presented below are made with the best intentions, aiming to contribute to the enhancement of this manuscript.
The review covers a wide range of topics related to amphibian conservation and Reproductive Biotechnologies, Conservation Breeding Programs, and Biobanking (RBCs), but lacks depth in certain areas, such as the challenges of practical implementation of RBCs, and particularly due to the superficiality in the section "3. Reproductive Biotechnologies". The relevance of the review is high, given the urgent need for conservation strategies, but could benefit from providing more comprehensive information, including the concept of One Conservation, which also uses reproductive biotechnologies as a conservation tool. The text is often redundant and repetitive, with repetition of ideas in some parts that could be condensed. Reorganize sections to improve logical flow, particularly in the discussion of reproductive biotechnologies and their applications.
This reviewer appreciates the comprehensive nature of the manuscript but suggests several improvements to enhance its clarity, accuracy, and overall impact. The following points should be addressed:
A. Terminology and Consistency:
- Ensure consistent use of correct technical terms throughout the manuscript, as detailed in the following comments.
- Pay particular attention to the use of "reproductive biotechnologies", "germplasm biobanking", and "genetic diversity".
B. Data Updates:
- Update statistical information, particularly regarding species numbers and conservation statuses, to reflect the most current data available from IUCN and AmphibiaWeb.
C. Conceptual Framework:
- Incorporate also the One Conservation concept, which is important for understanding the holistic approach to species conservation, including the role of reproductive biotechnologies.
D. Methodological Clarity / Technical Details:
- Provide more detailed explanations of methodologies, especially regarding sperm collection techniques and their comparative stress levels on animals.
- Expand on the discussion of semen activation, storage, and quality assessment in amphibians, including relevant references and current best practices.
- Provide more information on the use of semen diluents and extenders for sperm maturation and storage.
- Clarify the rationale behind preferences for certain methods (e.g., testicular tissue extraction vs. hormonal stimulation) with supporting evidence.
E. Ethical Considerations:
- Re-evaluate and clarify statements regarding the stress levels of different procedures, ensuring that claims are well-supported by evidence.
Common technical terms in animal reproduction were not adequately employed, being replaced by awkward terms. Therefore, the following terms should be corrected throughout the article:
- "Reproductive biotechnologies" instead of "reproduction biotechnologies" - because "reproductive" refers directly to processes related to reproduction, while "reproduction" is more often used as a noun describing the act or process of reproducing. Similarly, change "advanced reproduction technologies" to "advanced reproductive technologies"
- "Germplasm biobanking" instead of just "biobanking" - because the authors are working with reproductive cells and not biobanks of diverse materials (e.g., blood, mucus, feces, etc.)
- "Genetic diversity" instead of "genotypic variation". This is because "genetic diversity" refers to the variety of genes and alleles within a population or species, being a broader concept that includes both genetic variability and genotypic diversity, and is crucial for the adaptation and survival of species. The term "genotypic variation" refers only to differences in genotypes between individuals in a population.
- Use "chilled" and not "refrigerated" for sperm and oocytes. Using "chilled" instead of "refrigerated" when referring to sperm storage emphasizes the importance of maintaining a consistent and ideal temperature range to effectively preserve sperm function and viability.
- "Insemination with cryopreserved sperm" vs "fertilisation with cryopreserved sperm" - Fertilisation is correct when used in species with external fertilization, like most amphibians. However, some have internal fertilization, although without copulation. It would be interesting for the authors to bring to light that most amphibians have external fertilization and some may have internal, but without copulation. In animal reproduction, fertilization refers to the process of fertilization where sperm are used to fertilize an oocyte outside the body, usually in a controlled laboratory environment (or in the case of most amphibians). Insemination refers to the introduction of sperm directly into the female reproductive tract, more commonly through artificial insemination, and in species with internal fertilization.
3. Specific Comments
Lines 76-77 - It is necessary to update the data consulted a year ago. As of the present date, the IUCN accesses 8011 species, with 36.4% (2912) having some degree of threat of extinction, 52.3% NT or LC, and 11.3% (908) species with Data Deficient. AmphibiaWeb currently reports 8772 species.
Line 98 - Update data. Currently in the IUCN, there are 799 critically endangered and 1263 endangered species.
Line 102 - Suggestions for complementary references for wildfires: https://doi.org/10.1016/j.foreco.2023.121556 http://dx.doi.org/10.1163/15685381-bja10161 https://doi.org/10.1038/s41598-021-02844-5
Lines 121-132 - especially between 130-132 - The authors should include the One Conservation concept, being the first to emphatically bring to light the need for the development of reproductive biotechnologies in species conservation. https://doi.org/10.1590/1984-3143-AR2021-0024
Line 126 - First time the abbreviation CBPs appears, however, the authors do not clarify this abbreviation. This reviewer understands that it should be "conservation breeding programs", however on line 133, the authors use the abbreviation RBC (and not CBP).
Lines 137-138 - It is far beyond integrating in situ and ex situ. Rather, it is about bringing all layers of society as agents of conservation. See the One Conservation concept.
Lines 143-144 - The One Conservation concept citation should also be included here.
Lines 157-159 - It is not possible to understand what the authors wish to convey in this sentence.
Lines 167-168 - The sentence is not clear and seems to say that cultural factors are playing a role in the widespread adoption of RBCs.
Lines 187-188 - Also framed within the One Conservation concept, which should be utilized.
Lines 286-289 - Although this reviewer agrees that the main objective for biobanking should be Critically Endangered and Endangered species, any opportunity for producing biobank material for Vulnerable species should be taken advantage of. This is because today, they are classified only as Vulnerable. However, it is possible that when the degree of threat increases, it may be too late to form a biobank rich in genetic diversity. Therefore, despite not being objectives of efforts, the opportunistic collection of genetic material in an in situ environment is also important for vulnerable species. It would be important for the authors to consider this point of view.
Line 291 - "Sperm production" OR "sperm count"... and not "sperm yield". This Reviewer did not understand to what the authors applied the term yield.
Line 298 - Correct to: "was found by Watt et al [140]", to comply with the journal's standards.
Lines 438-441 - This Reviewer could not understand what the authors intended about vouchering.... What did they mean when they used the term "sources of biobanked sperm"? Even less did this reviewer understand the relationship between vouchering and reproductive biotechnologies.
Lines 464 to 481 - It would be good to rewrite these two paragraphs, making them clearer and with a more logical sequence of information. It was confusing and there seems to be information out of a clear order.
Lines 466-467 - Why is it an unfounded bureaucratic impediment? This Reviewer assumes that for the recovery of sperm from testicular tissues, euthanasia is necessary. However, the authors did not make it clear whether this form is the only possible one in certain species, or if spermiation is possible without euthanasia. Justify why they comment it is "unfounded", as the following paragraph did not convince this reviewer.
Lines 482-492 - Discussion of hormonal stimulation options used in amphibians was missing.
Lines 483-484 - Please provide references for the statement "but generally requires injection, and is a more stressful and less efficient alternative to sperm collection from testicular tissue". This reviewer could not understand how the application of injection can be more stressful than euthanasia...
Lines 485-487 - The statement that "hormonal stimulation only yields low sperm numbers, and concentrations, of inferior sperm quality ... limiting its utility" is very strong, requiring studies with different hormonal stimulation protocols in the species to really maintain this assertion. The presented references do not support this conclusion.
Line 513 - The term "exceptionally stressful" seems to be an opinion of the authors, as the mentioned article does not mention stress. Apparently, there is a bias of the authors towards the use of injection and preference for euthanasia. In contrast, the authors report on line 530 that ovarian excision is a low-stress technique. This Reviewer cannot understand how ovarian excision is performed with low stress and an injection causes high stress.
See also Lines 785-786 for the same issue.
Line 538 - "Sperm Qualities" is very awkward. Change to "Semen Quality".
Lines 544-547 - The authors use awkward and unusual terms in animal reproduction throughout this section. Instead of "subjectively assessed by observation", the correct term would be "subjectively assessed by eyes". Regarding CASA, the authors need to include the importance of specific setup for amphibians and even between species with significant sperm differences. This article, although on elasmobranchs, clearly brings this importance and should be used by the authors: https://doi.org/10.1016/j.therwi.2024.100091
Lines 538-551 - There is no comment on the activation of amphibian semen... Suggestions: https://doi.org/10.1111/jeb.12584 and https://doi.org/10.1016/j.theriogenology.2014.09.018 (this one from the same first author of this manuscript).
Line 552 - The term "Refrigerated Storage of Sperm" is incorrect and should be replaced with "Chilled Sperm Storage". This section also has awkward terms and should be revised. "Chilled semen" and not "refrigerated sperm".
Lines 552-563 - There is no mention of the use of semen diluents/extenders for sperm maturation of semen obtained from testicular tissues... And it also does not mention types of diluents for maintaining chilled semen.
Line 627 - The use of "miniscule amount of egg yolk" for mammals sounds awkward. Simply change to "minimal yolk content". This is because mammalian oocytes are "vitellins", and therefore lack a typical yolk.
Line 646 - "Artificial insemination" cannot be encompassed within "artificial fertilization". Artificial insemination refers to the introduction of sperm directly into the female reproductive tract. Fertilization refers to the process of fertilization where sperm fertilize an oocyte outside the body.
Item 5. Ethics and Communication - The text seems to favor the collection of sperm from testicular tissue over hormonal stimulation, which may reflect a particular bias. This was also noticed earlier by This Reviewer. Additionally, the discussion on speciesism and the selection of target species based on political or institutional foundations reflects a particular perspective of the authors on conservation, which may not be very accurate. This Reviewer suggests a revision of the sentence, making it evident that it is a point of view of the authors.
Lines 749-750 - Just a comment. Not only in amphibians, but in animal reproduction in general, there are few publications on ethical standards.
Item 6. Current and Future Application - This Reviewer commends the authors for this discussion and shares a similar view on the topic.
Missing Conclusions. According to the Instructions for Authors, this section is mandatory, with one or two paragraphs to end the main text.
Comments on the Quality of English LanguageThe manuscript is understandable and well-structured, but it would benefit from moderate editing to improve clarity. Some sentences are overly complex and could be divided to enhance readability. Additionally, certain grammatical constructions need adjustment. There is some repetition of ideas that could be condensed to avoid redundancy. Some terms should be reviewed to ensure they are up-to-date with those used in animal reproduction and more recent scientific literature.
Author Response
We thank the reviewer for very perceptive and constructive comments that have led to a much improved review. We have attached a doc that describes how we addressed the comments. In addition we have presented the editors with a track changed document with the changes by both referee 2 and 3, and a clean copy.

Reviewer 3 Report
Comments and Suggestions for Authors
In addition to my general remarks attached, the authors should address the projected increases in the release of greenhouse gases, endocrine-disrupting chemicals, and micro/nano plastic pollution (including the releases of toxic chemicals, such as mercury and iron, from melting glaciers and polar ice related to climate change) that will likely be occurring at a much faster rate than can be offset by the slow processes of artificially augmented natural reproduction. And even if this approach were to succeed for amphibians, what about the extinction rates of so many other vertebrate, invertebrate, and plant species?

My detailed comments on the quality of English communication can be found in the attachment.
Author Response
We thank the reviewer for very perceptive and constructive comments that have led to a much improved review. We have accepted all the changes presented on referee3's PDF as included in a track changed document with the changes by both referee 2 and 3. We have presented the editors with a clean copy.

Round 2
Reviewer 2 Report
Comments and Suggestions for Authors
This reviewer commends the authors for the excellent revisions made. The article is undoubtedly unique, compiling all fundamental aspects related to Reproduction and Advanced Biotechnologies, Biobanking of Germplasm and Somatic Cells, and Conservation Breeding Programs (RBCs).
The authors have successfully addressed the recommendations or provided well-founded and courteous justifications for those not fully incorporated. Whether or not this reviewer agrees with these justifications, the intention was solely to enrich the manuscript without impacting the authors' voice. Therefore, this reviewer is satisfied with the explanations provided.
The revised version is fantastic, with improved text fluidity and enriched content. It allows those unfamiliar with amphibian reproduction to grasp the subject without detracting from the scientific depth for specialists in this field. Reviewing this version was not only enriching but also enjoyable.
The article is outstanding, and this reviewer will undoubtedly consider it essential reading for future work on amphibian reproduction. Once again, this reviewer congratulates the authors on an exemplary article.